# A Classification of Elements of Function Space $F(\mathbb{R},\mathbb{R})$

Mohsen Soltanifar [1,2,3]

1    Analytics Division, College of Professional Studies, Northeastern University, 1400-410 West Georgia Street, Vancouver, BC V6B 1Z3, Canada; mohsen.soltanifar@alum.utoronto.ca
2    Biostatistics Division, Dalla Lana School of Public Health, University of Toronto, 620-155 College Street, Toronto, ON M5T 3M7, Canada
3    Biostatistics & Programming Division, Biometrics Department, ClinChoice Inc., 750-2 Robert Speck Parkway, Mississauga, ON L4Z 1H8, Canada

**Abstract:** In this paper, we present a constructive description of the function space of all real-valued functions on $\mathbb{R}$ ($F(\mathbb{R},\mathbb{R})$) by presenting a partition of it into 28 distinct blocks and a closed-form formula for the representative function of each of them. Each block contains elements that share common features in terms of the cardinality of their sets of continuity and differentiability. Alongside this classification, we introduce the concept of the *Connection*, which reveals a special relationship structure between the well-known representatives of four of the blocks: the Cantor function, the Dirichlet function, the Thomae function, and the Weierstrass function. Despite the significance of this field, several perspectives remain unexplored.

**Keywords:** real-valued functions; cardinals; Cantor function; Thomae function; Weierstrass function; Dirichlet function; partition

**MSC:** 05A18; 26A15; 26A21; 26A24; 26A30; 03E10

> *From the paradise, that Cantor created for us, no-one shall be able to expel us.*
> (David Hilbert-1925)

## 1. Introduction

### 1.1. Real Valued Functions

The theory of functions of a real variable was first treated by the Italian mathematician Ulisse Dini in 1878 [1]. This theory was constructed through the enlargement and deepening of set theory and was later developed separately and in parallel with its mentioned parent theory [1]. The development of this theory can be divided into three periods: the first period (1867–1902) saw extensive investigation into various topics of classical analysis, such as integrals, derivatives, and point set theory; the second period (1902–1930) marked the solidification of the theory of functions of a real variable as an independent mathematical discipline; and the third period (1930-present) is characterized by the study of the theory of real-valued functions in connection with functional analysis [1].

### 1.2. Motivation

The function space of all real-valued functions on the real line ($F(\mathbb{R},\mathbb{R})$) is an infinite-dimensional vector space with a nonconstructive basis. It encompasses various types of functions with pathological and chaotic structures. Here, we refer to the term "pathological" in the sense of contradicting features with perceived human intuition. Furthermore, we refer to the term "chaotic" in the context of dynamical systems where, at any neighborhood of a given point in the function domain, there is an unpredictable change in the values of the function [2]. As researchers' attention has shifted from pure existential mathematics to constructive mathematics [3,4], mathematicians have made numerous attempts to





focus on special subsets of this vast vector space (e.g., all real-valued continuous functions [5]) or to classify this vector space using novel ancillary concepts. Examples of such classifications include those based on (i) set theory (surjective, injective, bijective, etc.), (ii) operators (additive, multiplicative, even, odd, etc.), (iii) topology (continuous, open, closed, etc.), (iv) properties concerning real numbers (differentiable, smooth, convex, etc.), and (v) measurability (Borel status, Baire status, etc.) [6–8].

This work presents another attempt to describe the vector space of real-valued functions on the real line from a *constructive mathematics* perspective. Specifically, it aims to classify the elements of this vector space using their associated information on cardinality, continuity, and differentiability. Furthermore, based on this classification, it establishes a particular relationship among four of them: the Cantor function, the Dirichlet function, the Thomae function, and the Weierstrass function.

### 1.3. Study Outline

This paper is divided into four sections. The first section provides the necessary preliminaries in set theory, linear algebra, and special functions, which are essential for the following sections. In the second section, we discuss the partition of $F(\mathbb{R}, \mathbb{R})$ into 28 blocks of functions, presenting constructive examples for each block. The third section focuses on the relationship between these 28 blocks of functions using graph theory, with a specific emphasis on the four main plausible functions. Finally, we conclude the work with a discussion section on the current results and future directions.

## 2. Preliminaries

Readers who have studied the key topics of Analysis and Linear Algebra are well-equipped with the following notations, definitions, and results in the areas of "Set Theory" [9–11], "Linear Algebra" [9,12–14], and "Special Functions" [15–22].

### 2.1. Set Theory

**Proposition 1.** *Let $\mathbb{R}$ denote the set of real numbers and $A \subseteq \mathbb{R}$. Then, $A$ is empty, non-empty finite, denumerable, or uncountable. In these cases, we denote the cardinality of $A$ by $0, n(n \in \mathbb{N}), \aleph_0$, or $c$, respectively.*

**Remark 1.** *Henceforth, we assume there are only four types of subsets in the real line given Proposition 1, considering all non-empty finite sets of one category with common symbol "$n$" as their cardinal number.*

**Proposition 2.** *(i) Let $A \subseteq \mathbb{R}$ be uncountable and $A = A_1 \dot\cup A_2$. Then, $A_1$ or $A_2$ is uncountable. (ii) Let $A \subseteq B \subseteq \mathbb{R}$. Then, $0 \leq Card(A) \leq Card(B) \leq c$.*

**Proposition 3.** *Let $C$ be the ternary Cantor set, i.e., $C = \{x \in [0,1] | x := \sum_{n=1}^{\infty} \frac{a_n}{3^n} : a_n = 0, 2(n \in \mathbb{N})\}$. Then, one can write $C = \cap_{n=1}^{\infty} C_n$ where $C_n$ is the disjoint union of $2^n$ intervals of the form $I_{n,k} := [a_{n,k}, b_{n,k}]$ $(1 \leq k \leq n)$ each of the length $3^{-n}, (n \geq 1)$.*

**Remark 2.** *Given $\mathbb{C}_{unc} = \{A \subseteq C | Card(A) = c\}$, we have $Card(\mathbb{C}_{unc}) = 2^c$.*

### 2.2. Linear Algebra

**Definition 1.** *Let $\mathbb{R}$ denote the set of real numbers. We define (i) $F(\mathbb{R}, \mathbb{R}) = \{f : \mathbb{R} \to \mathbb{R} | f \text{ is a function}\}$; (ii) $C(\mathbb{R}, \mathbb{R}) = \{f : \mathbb{R} \to \mathbb{R} | f \text{ is a continuous function everywhere}\}$; and (iii) $D(\mathbb{R}, \mathbb{R}) = \{f : \mathbb{R} \to \mathbb{R} | f \text{ is a differentiable function everywhere}\}$.*

**Remark 3.** *The function space $F(\mathbb{R}, \mathbb{R})$ equipped with conventional addition $+$ and scalar multiplication $\bullet$ constitutes the vector space $(F(\mathbb{R}, \mathbb{R}), \mathbb{R}, +, \bullet)$. In addition, the vector spaces $(C(\mathbb{R}, \mathbb{R}), \mathbb{R}, +, \bullet)$, and $(D(\mathbb{R}, \mathbb{R}), \mathbb{R}, +, \bullet)$ are its sub-spaces.*

**Proposition 4.** *Given the sets introduced in Definition 1, we have (i) $D(\mathbb{R}, \mathbb{R}) \subsetneq C(\mathbb{R}, \mathbb{R}) \subsetneq F(\mathbb{R}, \mathbb{R})$; (ii) $Card(F(\mathbb{R}, \mathbb{R})) = 2^c$; (iii) $Card(C(\mathbb{R}, \mathbb{R})) = c$; and (iv) $Card(D(\mathbb{R}, \mathbb{R})) = c$.*

**Proposition 5.** $\dim(F(\mathbb{R}, \mathbb{R})) = 2^c$.

*2.3. Special Functions*

**Definition 2.** *Given indicator function $1_A (A \subseteq \mathbb{R})$, and the ternary Cantor set C. Then, the Cantor function $C(.)$, the Dirichlet function $D(.)$, the Thomae function $T(.)$, and the Weierstrass function $W(.)$ are defined on a closed-unit interval as:*

$$C(x := \sum_{n=1}^{\infty} \frac{a_{n,x}}{3^n}) = \frac{1}{2^{N_x}} + \sum_{n=1}^{N_x - 1} \frac{a_{n,x}}{2^{n+1}} : N_x = \min\{n \in \mathbb{N} : a_{n,x} = 1\}, \quad (1)$$

$$D(x) = 1_{\mathbb{Q}}(x), \quad (2)$$

$$T(\frac{m}{n} 1_{\mathbb{Q}}(x := \frac{m}{n}) + x 1_{\mathbb{Q}^c}(x)) = \frac{1}{n} 1_{\mathbb{Q}}(x) : (m, n) = 1, \quad (3)$$

$$W(x) = \sum_{n=0}^{\infty} \frac{\cos(21^n \pi x)}{3^n}. \quad (4)$$

**Remark 4.** *We note that the definitions of the above functions have straightforward extension from the closed unit interval to the entire real line. Also, the introduced Weierstrass function here is a special case of the general form for $a = \frac{1}{3}$, $b = 21$.*

**Definition 3.** *Let C be the ternary Cantor set, $D(.)$ be the Dirichlet function, and $-\infty < a < b < \infty$. Then, for the triangular function $T^{(1)}(.)$ given by $T^{(1)}_{a,b}(x) = \sqrt{3}\left(\frac{b-a}{2} - |x - \frac{b+a}{2}|\right) 1_{[a,b]}(x)$, and the transformed cosine function $T^{(2)}(.)$ given by $T^{(2)}_{a,b}(x) = (b-a) * (1 - \cos(2\pi(\frac{x-a}{b-a}))) 1_{[a,b]}(x)$, we define two functions $f_C$ and $g_C$ on the real line as*

$$f_C(x) = (\sum_{n=1}^{\infty} f_n(x))D(x) : f_n(x) = \sum_{k=1}^{2^n} T^{(1)}_{a_{n,k},b_{n,k}}(x) 1_{[a_{n,k},b_{n,k}]}(x), \ (n \geq 1) \quad (5)$$

$$g_C(x) = (\sum_{n=1}^{\infty} g_n(x))D(x) : g_n(x) = \sum_{k=1}^{2^n} T^{(2)}_{a_{n,k},b_{n,k}}(x) 1_{[a_{n,k},b_{n,k}]}(x) \ (n \geq 1). \quad (6)$$

**Remark 5.** *While the triangular function is continuous everywhere and has no derivative at points $x = a, b$, the linear transformed cosine function is differentiable everywhere and, in particular, its derivative at points $x = a, b$ is zero. These properties are inherited by the associated functions $f_C$ and $g_C$, respectively.*

## 3. Main Results

*3.1. Partition of $F(\mathbb{R}, \mathbb{R})$ with Scenario Classification & Examples*

We present a constructive perspective of the function space $F(\mathbb{R}, \mathbb{R})$ by its blockization in three parts: (i) Existence of the Blocks, (ii) Construction of the Blocks, and, (iii) Size of the Blocks.

### 3.1.1. Existence of the Blocks

We start by partitioning the function space $F(\mathbb{R}, \mathbb{R})$ into a finite number of blocks using the intersectionality of three concepts: "cardinal number", "continuity", and "differentiability". Here, two real-valued functions defined on $\mathbb{R}$ belong to the same block if and only if their associated set of continuities, discontinuities, differentiabilities, and non-differentiabilities have the same cardinal number, explained in detail below.

**Definition 4.** *Let $f, g \in F(\mathbb{R}, \mathbb{R})$, with associated sets of continuities $C_f, C_g$ and the associated sets of differentiabilities $D_f, D_g$. Then, $f$ is equivalent to $g$ in structure, denoted by $f \mathcal{R}_s g$, whenever (i) $Card(C_f) = Card(C_g)$; (ii) $Card(C_f^c) = Card(C_g^c)$; (iii) $Card(D_f) = Card(D_g)$; and (iv) $Card(D_f^c) = Card(D_g^c)$.*

It is trivial that $\mathcal{R}_s$ in Definition 4 is an equivalence relation on $F(\mathbb{R}, \mathbb{R})$. Hence, it induces a partition on it as follows.

**Theorem 1.** *The function space $F(\mathbb{R}, \mathbb{R})$ may be partitioned into 28 unique distinct blocks:*

$$F(\mathbb{R}, \mathbb{R}) = \dot{\bigcup}_{i=1}^{28} [f_i], \tag{7}$$

$$[f_i] = \{f \in F(\mathbb{R}, \mathbb{R}) | f \mathcal{R}_s f_i\} (1 \le i \le 28). \tag{8}$$

**Proof.** First, let $f \in F(\mathbb{R}, \mathbb{R})$, and consider its set of continuity points $C_f$. Then, given $\mathbb{R} = C_f \dot{\cup} C_f^c$, by an application of Proposition 2 (i), it follows that at least $C_f$ or $C_f^c$ is uncountable. Subsequently, by Proposition 2 (ii), there are seven different scenarios for the cardinality of the pair $(C_f, C_f^c)$, including $(0, c), (n, c), (\aleph_0, c), (c, c), (c, \aleph_0), (c, n)$, or $(c, 0)$. A similar argument for the set of differentiabilities $D_f$ with the same seven blocks holds. Secondly, by multiplication principle, it appears that there are $7 \times 7 = 49$ blocks of functions. However, by two applications of Proposition 2 (ii) for $A = D_f$ and $B = C_f$, and for $A = C_f^c$ and $B = D_f^c$, only $(1 + 2 + \cdots + 7) = 28$ blocks exist. Table 1 lists these blocks.  $\square$

**Table 1.** List of 28 representatives blocks of partition of $F(\mathbb{R}, \mathbb{R})$.

|  |  | Continuity | | Differentiability | |
| --- | --- | --- | --- | --- | --- |
| # | Case | $Card(C_f)$ | $Card(C_f^c)$ | $Card(D_f)$ | $Card(D_f^c)$ |
| 1 | 1-1 | 0 | $c$ | 0 | $c$ |
| 2 | 2-1 | $n$ | $c$ | 0 | $c$ |
| 3 | 2-2 | — | — | $n$ | $c$ |
| 4 | 3-1 | $\aleph_0$ | $c$ | 0 | $c$ |
| 5 | 3-2 | — | — | $n$ | $c$ |
| 6 | 3-3 | — | — | $\aleph_0$ | $c$ |
| 7 | 4-1 | $c$ | $c$ | 0 | $c$ |
| 8 | 4-2 | — | — | $n$ | $c$ |
| 9 | 4-3 | — | — | $\aleph_0$ | $c$ |
| 10 | 4-4 | — | — | $c$ | $c$ |
| 11 | 5-1 | $c$ | $\aleph_0$ | 0 | $c$ |
| 12 | 5-2 | — | — | $n$ | $c$ |
| 13 | 5-3 | — | — | $\aleph_0$ | $c$ |
| 14 | 5-4 | — | — | $c$ | $c$ |
| 15 | 5-5 | — | — | $c$ | $\aleph_0$ |
| 16 | 6-1 | $c$ | $n$ | 0 | $c$ |
| 17 | 6-2 | — | — | $n$ | $c$ |
| 18 | 6-3 | — | — | $\aleph_0$ | $c$ |
| 19 | 6-4 | — | — | $c$ | $c$ |
| 20 | 6-5 | — | — | $c$ | $\aleph_0$ |
| 21 | 6-6 | — | — | $c$ | $n$ |
| 22 | 7-1 | $c$ | 0 | 0 | $c$ |
| 23 | 7-2 | — | — | $n$ | $c$ |
| 24 | 7-3 | — | — | $\aleph_0$ | $c$ |
| 25 | 7-4 | — | — | $c$ | $c$ |
| 26 | 7-5 | — | — | $c$ | $\aleph_0$ |
| 27 | 7-6 | — | — | $c$ | $n$ |
| 28 | 7-7 | — | — | $c$ | 0 |

**Corollary 1.** *Each of the blocks of functions with the most chaotic structure ($[f_1]$) and functions of the least chaotic structure ($[f_{28}] = D(\mathbb{R}, \mathbb{R})$) constitutes only 3.5% (1/28) of all blocks. Hence, 93% of blocks of functions fall between these two opposite extremes. Furthermore, the vector space of everywhere-continuous functions on the real line ($C(\mathbb{R}, \mathbb{R})$) constitutes 25% (7/28) of all blocks.*

3.1.2. Construction of the Blocks

We now consider the proposed blocks in Theorem 1 and investigate the existence of at least one explicit closed form function as the representative of each block, as shown below.

**Theorem 2.** *There is at least one constructive example representing each of 28 unique distinct blocks of functions in $F(\mathbb{R}, \mathbb{R})$.*

**Proof.** We consider the *i*th ($1 \leq i \leq 28$) row in the Table 1 and find a closed-form representative function $f_i (1 \leq i \leq 28)$. Table 2 lists all the functions. It is straightforward for the reader to check that each function satisfies the associated block requirements. □

**Table 2.** List of 28 representative functions for each of the 28 blocks $F(\mathbb{R}, \mathbb{R})$.

| # | Case | Representative $f(x)$ | Comments | $Card([f])$ |
|---|------|----------------------|----------|-------------|
| 1 | 1-1 | $D(x)$ | Dirichlet Function | $2^c$ |
| 2 | 2-1 | $(\prod_{k=1}^n (x-k))D(x)$ | | $2^c$ |
| 3 | 2-2 | $(\prod_{k=1}^n (x-k)^2)D(x)$ | | $2^c$ |
| 4 | 3-1 | $\sin(\pi x)D(x)$ | | $2^c$ |
| 5 | 3-2 | $(\sin(\pi x)\prod_{k=1}^n (x-k))D(x)$ | | $2^c$ |
| 6 | 3-3 | $(\sin^2(\pi x))D(x)$ | | $2^c$ |
| 7 | 4-1 | $f_C(x)$ | $C_f = C$ | $2^c$ |
| 8 | 4-2 | $(\prod_{k=1}^n (x-\frac{1}{3^k})^2)f_C(x)$ | | $2^c$ |
| 9 | 4-3 | $(\sin^2(\frac{\pi}{x}))f_C(x)$ | | $2^c$ |
| 10 | 4-4 | $g_C(x)$ | $C_f = D_f = C$ | $2^c$ |
| 11 | 5-1 | $T(x)$ | Thomae Function | $c$ |
| 12 | 5-2 | $(\prod_{k=1}^n (x-k)^2)T(x)$ | | $c$ |
| 13 | 5-3 | $(\sin^2(\pi x))T(x)$ | | $c$ |
| 14 | 5-4 | $T(x)1_{[0,1]}(x)$ | | $c$ |
| 15 | 5-5 | $\sum_{n=1}^{+\infty} \pi^n 1_{\{\pi^n\}}(x)$ | | $c$ |
| 16 | 6-1 | $W(x) + \sum_{k=1}^n \pi^k 1_{\{\pi^k\}}(x)$ | | $c$ |
| 17 | 6-2 | $(\prod_{k=1}^n (x-k)^2)(W(x) + \sum_{k=1}^n \pi^k 1_{\{\pi^k\}}(x))$ | | $c$ |
| 18 | 6-3 | $(\sin^2(\pi x))(W(x) + \sum_{k=1}^n \pi^k 1_{\{\pi^k\}}(x))$ | | $c$ |
| 19 | 6-4 | $(W(x) + \sum_{k=1}^n \pi^{-k} 1_{\{\pi^{-k}\}}(x))1_{[0,1]}(x)$ | | $c$ |
| 20 | 6-5 | $|\sin(\pi x)| + \sum_{k=1}^n \pi^k 1_{\{\pi^k\}}(x)$ | | $c$ |
| 21 | 6-6 | $\sum_{k=1}^n \pi^k 1_{\{\pi^k\}}(x)$ | | $c$ |
| 22 | 7-1 | $W(x)$ | Weierstrass Function | $c$ |
| 23 | 7-2 | $(\prod_{k=1}^n (x-k)^2)W(x)$ | | $c$ |
| 24 | 7-3 | $(\sin^2(\pi x))W(x)$ | | $c$ |
| 25 | 7-4 | $C(x)$ | Cantor Function | $c$ |
| 26 | 7-5 | $|\sin(\pi x)|$ | | $c$ |
| 27 | 7-6 | $|\prod_{k=1}^n (x-k)|$ | | $c$ |
| 28 | 7-7 | $x$ | | $c$ |

**Remark 6.** *The block of functions with the least chaotic structure ($[f_{28}]$) represented by the identity function I in the Table 2 lists many well-known functions, including all polynomials $p_m(.)(m \geq 0)$, the trigonometric functions $\sin(.)$ and $\cos(.)$, the exponential function $\exp(.)$, and the Volterra's function [23].*

**Remark 7.** *The ternary Cantor set ($C$) appears in the construction of representatives of $17.9\%(5/28)$ of the blocks. This shows that the ternary Cantor set has remarkable presence in the representative blocks of the space of real-valued functions on the real line.*

**Remark 8.** *The four well-known functions $C(.), D(.), T(.),$ and $W(.)$ show up in 64.3%(18/28) of the blocks where, for given representative function $f_i = (h_{i1} + h_{i2})h_{i3}$, at least one of $h_{ij}(j = 1, 2, 3)$ is one of these four functions.*

3.1.3. Size of the Blocks

We now consider the the problem of cardinality calculation of each presented block in Theorem 1. In the spirit of having some hints on the their sizes from information presented in Proposition 4, we investigate this in two steps as follows.

**Lemma 1.** *Let $F_1(\mathbb{R}, \mathbb{R}) = \cup_{i=1}^{10}[f_i]$ and $F_2(\mathbb{R}, \mathbb{R}) = \cup_{i=11}^{28}[f_i]$. Then,*
*(i) $Card(F_1(\mathbb{R}, \mathbb{R})) = 2^c$*
*(ii) $Card(F_2(\mathbb{R}, \mathbb{R})) = c$.*

**Proof.** First, given $F(\mathbb{R}, \mathbb{R}) = F_1(\mathbb{R}, \mathbb{R}) \cup F_2(\mathbb{R}, \mathbb{R})$, we have $2^c = Card(F_1(\mathbb{R}, \mathbb{R})) + Card(F_2(\mathbb{R}, \mathbb{R}))$. Hence, proving claim (ii) yields proving claim (i). Secondly, to prove claim (ii), let $X_{count} \subset \mathbb{R}$ be countable. Then, given $Card(C(X_{count}^c, \mathbb{R})) = c$ [24], we have $Card(\cup_{X_{count} \subset \mathbb{R}} C(X_{count}^c, \mathbb{R})) = c$. Consequently, $Card(F(\mathbb{N}, \mathbb{R}) \times \cup_{X_{count} \subset \mathbb{R}} C(X_{count}^c, \mathbb{R})) = c$. Finally, using the recent result, it is sufficient to consider the 1-1 mapping:

$$\phi \quad : \quad F_2(\mathbb{R}, \mathbb{R}) \to (F(\mathbb{N}, \mathbb{R}) \times \cup_{X_{count} \subset \mathbb{R}} C(X_{count}^c, \mathbb{R}))$$
$$\phi(f) \quad = \quad (f|_{C_f^c}, f|_{C_f}).$$

This completes the proof. $\square$

**Theorem 3.**
$$Card([f_i]) \quad = \quad 1_{[1,10]}(i) \times 2^c + 1_{[11,28]}(i) \times c \quad (1 \le i \le 28). \tag{9}$$

**Proof.** First, let $f \in F(\mathbb{Q}^c, \mathbb{R}_0^-)$. Then, we can extend $f$ to $\mathbb{R}$ by $f(\mathbb{Q}) = \{1\}$. A straightforward verification shows that $f \in [f_1]$. Thus, given the 1-1 mapping

$$\psi_1 \quad : \quad F(\mathbb{Q}^c, \mathbb{R}_0^-) \to [f_1]$$
$$\psi_1(f) \quad = \quad [f_1],$$

we have $2^c = Card(F(\mathbb{Q}^c, \mathbb{R}_0^-)) \le Card([f_1]) \le Card(F(\mathbb{R}, \mathbb{R})) = 2^c$. This yields $Card([f_1]) = 2^c$. Next, let $f \in [f_1]$ and define $g_i(.)$ $(2 \le i \le 6)$ by $g_i(x) = (\prod_{k=1}^n(x - k)), (\prod_{k=1}^n(x - k)^2), \sin(\pi x), (\sin(\pi x) \prod_{k=1}^n(x - k)), (\sin^2(\pi x))$ for $(2 \le i \le 6)$,. Then, using the CDF of the normal distribution $\Phi$, we have $g_i \Phi(f) \in [f_i]$ for $(2 \le i \le 6)$, respectively. Thus, given the 1-1 mapping

$$\psi_i \quad : \quad [f_1] \to [f_i](2 \le i \le 6)$$
$$\psi_i(f) \quad = \quad g_i \Phi(f),$$

we have $2^c = Card([f_1]) \le Card([f_i]) \le Card(F(\mathbb{R}, \mathbb{R})) = 2^c$, implying $Card([f_i]) = 2^c$ for $(2 \le i \le 6)$.

Secondly, let $A \in \mathbb{C}_{unc}$ and consider the modified $f_i^A(.)$ $(7 \le i \le 10)$ in Table 2 by $f_i^A(x) = f_A(x), (\prod_{k=1}^n(x - 1/3^k))f_A(x), (\sin^2(\frac{\pi}{x}))f_A(x), g_A(x)$ for $(7 \le i \le 10)$. Thus, given the 1-1 mapping

$$\psi_i \quad : \quad \mathbb{C}_{unc} \to [f_i](7 \le i \le 10)$$
$$\psi_i(A) \quad = \quad f_i^A,$$

we have $2^c = Card(\mathbb{C}_{unc}) \le Card([f_i]) \le Card(F(\mathbb{R}, \mathbb{R})) = 2^c$, implying $Card([f_i]) = 2^c$ for $(7 \le i \le 10)$.

Thirdly, let $p \in \mathbb{R}^+$. Then, using function $f_i$ $(11 \le i \le 28)$ in the Table 2, we have $pf_i \in [f_i]$ $(11 \le i \le 28)$. Hence, given the 1-1 mapping

$$\psi_i \quad : \quad \mathbb{R}^+ \to [f_i](11 \le i \le 28)$$
$$\psi_i(p) \quad = \quad p.f_i,$$

we have $c = Card(\mathbb{R}^+) \le Card([f_i])$, for $(11 \le i \le 28)$, respectively. Next, given $[f_i] \subseteq F_2(\mathbb{R}, \mathbb{R})$, for $(11 \le i \le 28)$, and Lemma 1 (ii), we have $Card([f_i]) \le c$ for $(11 \le i \le 28)$.

Accordingly, by the last two inequalities on the cardinals, it follows that: $Card([f_i]) = c$ ($11 \leq i \leq 28$). In particular, for $[f_{28}] = D(\mathbb{R}, \mathbb{R})$, we have $Card([f_{28}]) = c$.

This completes the proof. $\square$

**Remark 9.** *The cardinality of* $35.7\%(10/28)$ *of the blocks is* $2^c$*, while that of* $64.3\%(18/28)$ *of blocks is c. This indicates that the cardinal number c has almost double frequency of that of cardinal number* $2^c$ *in representing the size of blocks of the space of real-valued functions on the real line.*

*3.2. The Connection between Real-Valued Functions on $\mathbb{R}$*

3.2.1. The Relationship between the 28 Representative Functions

In the previous section, we observed that any given function $f \in F(\mathbb{R}, \mathbb{R})$ belongs to one of the 28 blocks of its partitions. Now, one may wonder how to connect these key functions. Trivially, the equivalence relation induced by the aforementioned partition is unhelpful in this regard. Hence, we consider an alternative approach. We begin with a definition:

**Definition 5.** *Given two functions* $f_1, f_2 \in F(\mathbb{R}, \mathbb{R})$*, $f_1$ is called connected to $f_2$, denoted by* $f_1 \overset{conn}{\sim} f_2$*, whenever, for some function* $g \in F(\mathbb{R}, \mathbb{R})$*, we have* $f_1 g \in [f_2]$*.*

**Remark 10.** *Additional restrictions on g: When* $g = 1$ *in Definition* 5*,* $\overset{conn}{\sim}$ *is transformed into the induced equivalence relation by the above partition in the Theorem* 1*; Furthermore, if g is a positive everywhere-differentiable function on* $\mathbb{R}$*, the aforementioned relation becomes an equivalence relation.*

It is trivial that, in Theorem 1, $f_i \overset{conn}{\sim} f_{28}$ for all ($1 \leq i \leq 27$). Figure 1 presents these relationships. As shown, the block of everywhere-differentiable functions $[f_{28}]$ is the sink node with the highest in-degree connectivity among all blocks of functions.

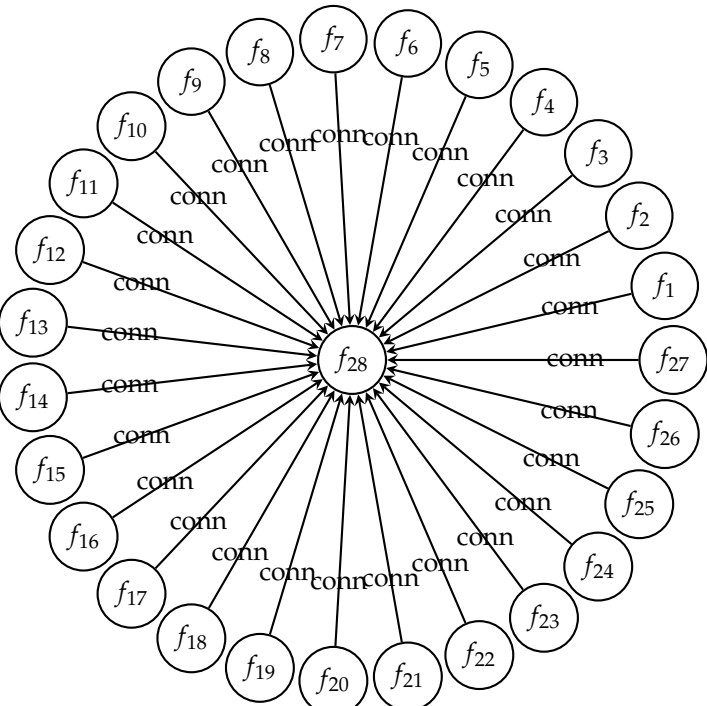

**Figure 1.** The (incomplete) graphical presentation of the relationship between all representatives of blocks of $F(\mathbb{R}, \mathbb{R})$ and $f_{28}$ the identity function.

### 3.2.2. The Relationship between the Big Four

As there are $4^{C_2^{28}} = 3.790327 \times 10^{227}$ potential scenarios for the complete graph in Figure 1, finding the relationships between all nodes of the graph appears to be a tedious and difficult task. However, we can identify the relationships between four of them, i.e., $f_1 = D, f_{11} = T, f_{22} = W$, and $f_{25} = C$, where there are only $4^{C_2^4} = 4096$ potential scenarios. Equipped with Definition 5, we have the following.

**Theorem 4.** *Given the above notations and definitions, we have (i) $W \overset{conn}{\sim} T$, (ii) $W \overset{conn}{\sim} C$, (iii) $W \overset{conn}{\sim} D$, (iv) $T \overset{conn}{\sim} D$, and (v) $C \overset{conn}{\sim} D$.*

**Proof.** It is sufficient for each case to present the $g$ function in Definition 5 as follows: (i) $g(x) = x + \sum_{n=1}^{+\infty} \pi^n 1_{\{\pi^n\}}(x)$; (ii) $g(x) = x(x-1)1_{[0,1]}(x)$; (iii)–(v) $g(x) = \sum_{-\infty}^{+\infty} 1_{A+2n}(x)$ : $A = (\sin(n))_{n=1}^{+\infty}$ dense in $[-1,1]$. $\quad\square$

Figure 2 presents a graphical overview of the results in Theorem 4. As is shown, the Weierstrass function (W) is the source-universal node with the highest out-degree connectivity; the Cantor function (C) and the Thomae function (T) are the bridging nodes; and the Dirichlet function (D) is the sink node with the highest in-degree connectivity.

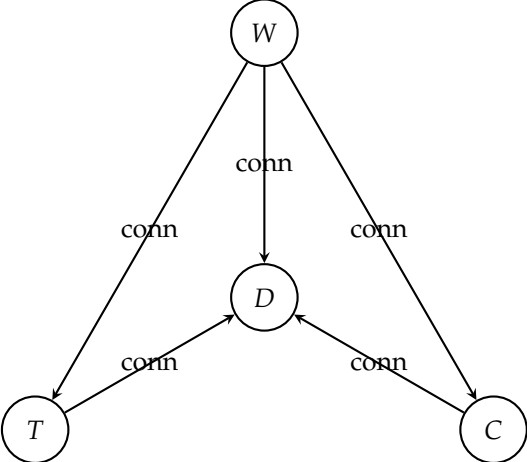

**Figure 2.** The graphical presentation of the relationship between well-known functions: the Cantor function (C), the Dirichlet function (D), the Thomae function (T), and the Weierstrass function (W).

## 4. Discussion

### 4.1. Summary & Contributions

This work presented a finite partition of the function spaces of all real-valued functions on $\mathbb{R}$ based on cardinality, continuity, and differentiability, along with constructive examples representing each block of the partition. In particular, it showed that the well-known Cantor function, the Dirichlet function, the Thomae function, and the Weierstrass function each represented a unique block of this partition. An additional aspect that adds more importance to these four functions is that they collectively appeared in the representation of almost two-thirds of the blocks. Furthermore, the concept of *Connection* among real-valued functions was introduced, and the unique connection relation between the mentioned functions was investigated.

Finally, this work's findings add more prominence to the Cantor set $C$ as well. While it had a remarkable presence in the construction of the representative functions of blocks, its size (e.g., Cardinal number $c$) had the highest presence in the set of sizes $\{c, 2^c\}$ of the representative blocks.

### 4.2. Limitations & Future Work

The limitations in this work are clear, and they open up new perspectives for further investigations. Firstly, we merged the cardinal number of all finite subsets of $\mathbb{R}$ with given symbol $n$. While this inaccuracy is a minimal price to pay for enabling the creation of the aforementioned finite partition, it should be noted. Secondly, the presented graph in Figure 1 needs to be completed for all its involved nodes. Thirdly, the *connection* relation in Definition 5 is not an equivalence, making it suboptimal. One open problem in this regard is investigating the results in Figures 1 and 2 when considering the equivalence relations mentioned in Remark 10. Finally, it is worth exploring how the equivalence relation in Definition 4 influences the structure and the number of blocks in the presented partition. When one presents a new equivalence relation on the function space $F(\mathbb{R}, \mathbb{R})$ by replacing some key features in the Definition 4 (such as continuity and differentiability) with other properties of real-valued functions on the real line (such as integrability, measurability, etc.), the structure and number of the blocks of the presented partition may change. This modification opens doors to many other uncultivated areas of partition representation of the function space $F(\mathbb{R}, \mathbb{R})$.

### 4.3. Conclusions

This work presented a constructive and finite partition-based description of the function space of all real-valued functions on $\mathbb{R}$ which has formerly been characterized by its pure existential infinite dimensional base. Additionally, it presented the concept of the "Connection" between elements of this space.

**Funding:** This research received no external funding.

**Institutional Review Board Statement:** Not applicable.

**Informed Consent Statement:** Not applicable.

**Data Availability Statement:** Not applicable.

**Acknowledgments:** The author is grateful to the journal reviewers for their constructive comments and suggestions on the first draft of the manuscript.

**Conflicts of Interest:** The author declares no conflict of interest.

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
