# Peer review of "A Classification of Elements of Function Space F( , )"

_mathematics, doi:10.3390/math11173715_

Round 1
Reviewer 1 Report
Authors can provide a clear explanation or a set of criteria for why these specific functions are chosen as representative for each block. If there is a shared intrinsic property, detailing it would be of benefit to the reader.
The authors can provide an additional section that speculates on potential changes in the function space classification when considering equivalence relations or other properties. This doesn't need to be a fully fleshed-out analysis but some educated guesses or hypotheses based on the authors' knowledge of the field. This can open the door for future research and spark interest in other researchers to explore these areas.
-
Reviewer 2 Report
The paper entitled "A classification of elements of function space F(RR) was considered by the author.
General Comment:
It was observed from the manuscript that some sections of the work were highlighted which the fact may have been submitted to a journal before submitting it here. I think the author should explain the reason for that.
Under the introduction and line 3, ....in parallel..(with what?), I think there should be a phrase to complete that sentence.
Under Remark 2.13, and line 3, ...particular 'it' should be "its"
Under the proof of Theorem 3.1 and line 2, Cf of Ccf 'in' the in should be is.
The word 'second' should be checked, at times it is good to say 'secondly'.
Finally, the paper seems in part as a review and survey, though the beauty of the work is that it provides an established intra-relationship between various real functions and some special functions. The paper opens a channel for further research in this direction
The language is ok except for few correction
Reviewer 3 Report
Dear Authors,
The introduction and abstract offer a nice introduction to the topic of the paper.
Remarks:
The cited references for special functions are not special function books or papers, therefore it must be revisited. Proposition 2.9 must be given a reference.
The Authors must define what do they mean by "chaotic" structure. Also it would be nice to explain how table 2 is made.
Authors must also cite some newer references related To the work.
The paper must be revised regarding the English language.
Detailed:
1.The paper concerns itself with obtaining unique classes of mappings defined on R which can be partitioned on R. The paper concerns itself with continuity and differentiability on the domain R of a function.
2. The strength of the paper is the topic which is being investigated which is indeed interesting and can bring more understanding to the relationship between mappings on the domain R which concern differentiability and continuity.
3. English language must be revised. An explanation must be given as to how table 2 was made. Explanation throughout the paper is missing. More attention must be put into the proofs. Citations which are given for Special functions are not valid. As there is no book cited related to special function theory.
All the best
The English must be revised.
Reviewer 4 Report
See pdf file

Round 2
Reviewer 1 Report
The authors have addressed all my concerns, Thanks
-
Reviewer 3 Report
Everything has been taken care of.
I recommend the paper as it is.
Reviewer 4 Report
I have no any remarks